# Methods for Radiolabelling Nanoparticles: PET Use (Part 2)

**DOI:** 10.3390/biom12101517

**Published:** 2022-10-20

**Authors:** Valeria Bentivoglio, Michela Varani, Chiara Lauri, Danilo Ranieri, Alberto Signore

**Affiliations:** 1Nuclear Medicine Unit, Department of Medical-Surgical Sciences and of Translational Medicine, Faculty of Medicine and Psychology, “Sapienza” University of Rome, 00185 Rome, Italy; 2Department of Clinical and Molecular Medicine, Faculty of Medicine and Psychology, “Sapienza” University of Rome, 00185 Rome, Italy

**Keywords:** nanoparticles, nanotechnology, nuclear medicine, radiolabelling, PET/CT

## Abstract

The use of radiolabelled nanoparticles (NPs) is a promising nuclear medicine tool for diagnostic and therapeutic purposes. Thanks to the heterogeneity of their material (organic or inorganic) and their unique physical and chemical characteristics, they are highly versatile for their use in several medical applications. In particular, they have shown interesting results as radiolabelled probes for positron emission tomography (PET) imaging. The high variability of NP types and the possibility to use several isotopes in the radiolabelling process implies different radiolabelling methods that have been applied over the previous years. In this review, we compare and summarize the different methods for NP radiolabelling with the most frequently used PET isotopes.

## 1. Introduction

Nuclear medicine is an important clinical field for the diagnosis and therapy of several diseases, especially in the oncological field.

Radionuclides can be linked to different molecules to perform a molecular imaging procedure or targeted radionuclide therapy, depending on their radioactive decay.

Nanoparticles (NPs) have emerged as a successful platform for drug delivery. Similarly to other compounds, NPs can be radiolabelled with diagnostic or therapeutic isotopes for different applications. Furthermore, they can be considered as theragnostic tools since the same NPs that are radiolabelled with different isotopes can be used for diagnostic purposes or for a therapeutic application [1,2].

For diagnostic imaging purpose, they can be used for positron emission tomography (PET) or single photon emission tomography (SPECT) depending on whether they are radiolabelled with a positron-emitting isotope or a gamma-ray emitting isotope, respectively.

NP radiolabelling can occur in different ways depending on the nature of the radioisotope and the type of NP that it is. In particular, the half-life of both of them is one of the most important parameters to take into consideration, particularly if one is using a positron-emitting isotope that has a high energy and a short half-life.

In the first part of the review, we analyzed the labelling of NPs with gamma-emitting isotopes, and here, we focus of the use of positron-emitting isotopes for diagnostic purposes with PET.

Generally, NPs can be labelled by “direct radiolabelling” when the radioisotope is bound to the surface or when it is encapsulated into the core of NPs or by “indirect labelling” when a chelator is being used to bridge the NP with the isotope.

In general, the first method has many advantages when it is compared to the second, such as the preservation of the nanomaterial structure and the reduction of the number of steps which makes this process less time-consuming.

There are many strategies to radiolabel NPs without bi-functional chelators (BFCs) that are specific to the nanomaterial and to the radioisotope.

For example, exclusive strategies for inorganic nanomaterial consists of mixing the radionuclide and the non-radioactive nanomaterial precursors, thereby obtaining a radiochemical doping of the NPs during the synthesis [3].

The radio-halogenation process is performed for the radiolabelling of several types of NPs that have tyrosine residues on their surface [4].

The chemical adsorption of the radionuclides is another commonly applied method based on the formation of the coordination bond between the chemical groups on the surface of the nanomaterial and the radionuclide. This method could be used with a variety of radionuclides, but a high temperature is required for it, thus making this method limited for heat-sensitive nanomaterials [5].

The direct radiolabelling method is also allowed by the physical interaction that occurs between the radionuclides and the nanomaterial, for example, ones that are based on electrostatic interactions. Nevertheless, these bonds are usually weak and, therefore, this method has not been extensively explored. The chemical characteristics of the isotope can also influence the method of radiolabelling. Indeed, the labelling of NPs by direct methods usually occurs with non-metallic radionuclides (e.g., fluorine-18, iodine-131, etc.).

Radionuclides with metallic properties (e.g., copper-64 and zirconium-89) often require a ligand system (chelating agent) that binds the radiometal ions in a stable complex [3]. The chelator can be acyclic or linear, such as deferoxamine (DFO), diethylenetriamine-N,N,N′,N,N-pentaacetic acid, pentetic acid, (Carboxymethyl)imino]bis(ethylenenitrilo)-tetra-acetic acid (DTPA) or Nitrilotriacetic acid (NTA), etc., or macrocyclic, such as 2,2′,2′′,2′′′-(1,4,7,10-Tetraazacyclododecane-1,4,7,10-tetrayl)tetraacetic acid (DOTA), 1,4,7-triazacyclononane-N,N′,N′′-triacetic acid (NOTA), 1,4,7-triazaciclononane, 1-glutaric-4,7-acetic acid (NODAGA), Triethylenetetramine (TETA), etc. [6].

In the choice of the chelating agent, it is important to consider the coordination number or the oxidation state of the radioisotope to achieve a degree of final thermodynamic stability. Linear chelating agents are generally less rigid in their structure than macrocyclics are, and therefore, they require milder temperatures and faster reaction times. Macrocyclic chelators, being a complex structure, require a higher temperature and slow binding kinetics, but they have a higher degree of final stability. This could be an important factor in the choice of the correct method that is used to radiolabel the NPs due to the possibility of the aggregation or degradation of the nanosuspensions during the chelating process [7].

The chelators that are used in the radiolabelling of the NPs are defined BFCs as they are characterized by a double function; one is able to bind the radioisotope, and one is able to bind the NPs through a functional group on their surface. In this way, the radiolabelling of the NPs occurs ‘indirectly’ [8]. The radiolabelling process using the BFCs can be obtained by using two different approaches: the BFC can be linked to the NPs, and then, to the radioisotope, or the BFC can be linked first to the radioisotope, and then, to the NPs (Figure 1).

## 2. Radiolabelled NPs for PET Imaging

The positron-emitting radionuclides that are used in nuclear medicine have usually a shorter half-life than the single-photon emitters do. The most frequently used radionuclides with a short half-life are gallium-68, fluorine-18, carbon-11, nitrogen-13, and oxygen-15, which have ranges that span from 122 s to 109.7 min. The half-lives of the longer-lived positron emitters are 12.701 h for copper-64, 78.4 h for zirconium-89, and 100.22 h for iodine-124, respectively. In the case of indirect labelling, the most frequently used chelators are DOTA and NOTA, or DTPA and DFO [9].

Most of the studies that are reported in the literature focus on the use of NPs that are radiolabelled with positron-emitting isotopes for tumor imaging, but they have also been used for identifying rheumatic, neurological, and cardiovascular diseases [10].

### 2.1. Radiolabelling with Copper-64

The use of ^64^Cu for the radiolabelling of NPs is raising interest in both the preclinical and the clinical field. Its long relative half-life allows one to study the biodistribution and tumor targeting of the radiolabelled NPs for up to 48 h [11]. The chemical properties of this radiometal allows the use of different chelators that can be conjugated to different molecules. However, the conjugation of them with the chelator could influence the properties of the NPs and reduce the capability of the specific targeting technique.

#### 2.1.1. Direct Radiolabelling

The direct labelling of the NPs with ^64^Cu can be obtained with those nanomaterials that are defined as electron donors that have a high affinity with those radioisotopes that are defined as electron acceptors.

^64^Cu^2+^ ions (3d^9^) require an electron to have a stable electronic configuration, and for this reason, it is easy to label it with the donor nanomaterials. Shi et al. employed graphene nanomaterials as electron donors for ^64^Cu, thereby performing a stable direct labelling procedure without the use of BFCs. They showed that the labelling procedure is influenced by the temperature of the reaction and the concentration of the NPs. The highest labelling efficiency (LE), 75.5 ± 1.7%, was obtained with a concentration of 0.5 mg/mL^−1^ at 75 °C after 60 min of incubation [12].

The same method was applied to radiolabel silica NPs (SNPs), which were synthetized with the incorporation of oxygen atoms that were arranged in symmetry to be the electron donors for ^64^Cu. The radiolabelling occurred by simply incubating the free radioisotope at 70 °C for 60 min, and there was a final radiochemical yield (RCY) of 99% after the centrifugation of it. The RCY improves with increasing temperatures (from 4 to 70 °C), but no correlation has been shown when one is varying the pH (5.7–8.8) [13].

Other silica NPs cannot bind ^64^Cu stably; they dissociate rapidly under the physiological [14].

Several other metal nanomaterials can be labelled with metallic radioisotope by following the same principle of chemical affinity. For instance, iron oxide nanoparticles (IONPs) and gold nanoparticles (AuNPs) have been successfully radiolabelled with ^64^Cu without the use of a BFC due to the favorable characteristics of the magnetic NPs. In particular, IONPs were radiolabelled by Boros et al. with a chelator-free approach that was defined as a heat-induced metal ion binding method, thereby avoiding the expected multi-step radiolabelling in the indirect radiolabelling procedure. They also demonstrated the versatility of this method, which can be used with other metal isotopes such as ^111^In^3+^ and ^89^Zr^4+^, with a final RCY that was between 66–93% and radiochemical purities that were more than 98% [15].

Sun et al. radiolabelled AuNPs by chemically reducing the radioisotope on the surface of the pegylated NPs. The protocol included a reduction of ^64^Cu by hydrazine (N_2_H_4_) in presence of amine-poly-ethylene-glycol-thiol (PEG) and poly(acrylic acid) (PAA) on the surface of the AuNPs. They found that the presence of N_2_H_4_ is needed for the efficiency of the radiolabelling, with them achieving a final RCY of 100% in the presence of the reducing agent in comparison to this being 30% without it [16]. Similarly, Fan et al. used the intrinsic ability of water-soluble melanin NPs (MNPs) to bind metal ions for their radiolabelling with ^64^Cu^2+^. Indeed, the method that was applied was a single-step procedure, wherein CuCl_2_ which was in a buffer solution (pH = 5.5) was incubated with the NPs for 1 h at 40 °C. The stability test of the radiolabelled NPs was performed in phosphate-buffered saline (PBS) at 37 °C, with only ∼3% ^64^Cu^2+^ being released from the MNPs after 24 h of incubation [17]. A cation exchange approach can also occur between ^64^Cu and quantum dots (QDs). The radiolabelling can be performed by adding ^64^CuCl_2_ into a NP solution at room temperature while it is magnetically stirred at 95 °C for 1 h with 100% of the isotope being incorporated [18].

Single-well carbon nanotubes (SWCNTs) were directly radiolabelled with ^64^Cu using a one-step procedure by incubating the isotope and the NPs under a sonication condition for 1 h. However, the stability of the radiopharmaceutical decreased up to 50% in the serum, thereby confirming the poor stability of this radiolabelling approach for SWCNTs [19].

#### 2.1.2. Radiolabelling with Bifunctional Chelators (BFCs)

DOTA is the most frequently used BFC for ^64^Cu labelling since after the complexation with Cu^2+^-ions, it forms a stable complex, thereby leaving two carboxylic functions that are free to conjugate with the NPs and other molecules. The most frequently used method radiolabelling of NPs with ^64^Cu is a post-synthesis process: the NPs are synthetized, coupled with the BFC, and the isotope is added at the end [20].

Recently, an efficient strategy to indirectly radiolabel NPs with ^64^Cu is with a procedure that is defined as “click chemistry” or an azide–alkyne cycloaddition strategy [21].

The procedure includes the pre-radiolabelling of the chelating agent with a high LE, which is followed by the conjugation of the radiolabelling complex with the NPs. Despite this method leading to a high LE and RCY, the conjugation of DOTA to the NPs and then the radiolabelling of the complex is the most frequently used procedure.

For labelling the BFC before its conjugation with the NPs, several steps have to be followed as in the case of glycol chitosan NPs (CNPs) that are radiolabelled with ^64^Cu via a click-chemistry procedure. In the first step, the azide (N_3_) group was incorporated to the CNPs, and after that, the strained cyclooctyne derivative, dibenzyl cyclooctyne (DBCO) that was conjugated with DOTA, was synthesized for preparing the pre-radiolabelled alkyne complex. Following their incubation with the ^64^Cu, the NPs showed a high LE and RCY (>98%). This is a very fast way to radiolabel the NPs, and it was accomplished within 30 min in aqueous conditions with great efficiency. In addition, this method did not show any significant effect on the physicochemical properties of the NPs. The same method was used by Zeng et al. for the radiolabelling of the core in shell-crosslinked NPs (SCK-NPs) with a high LE [22,23].

For labelling the NPs after their conjugation with a BFC, such as DOTA, the NPs should be first activated with a (1-ethyl-3-(3-dimethylaminopropyl)carbodiimide (EDC) and sulpho-N-hydroxysuccinamide (NHS) protocol, then they should be conjugated with DOTA, and finally, they should be radiolabelled with the isotope. Gadolinium vanadate (GdVO_4_) ultrathin nanosheets (NSs) were firstly activated with EDC/NHs in a PBS solution for 3 h, and then, they were conjugated with DOTA-NH_2_ while they were stirred for 3 h. At the end of the process, ^64^CuCl_2_ was added to this complex in 0.1 M sodium acetate (pH 5.5) at room temperature for 1.5 h [24].

A different approach that can be used is the activation of DOTA with EDC/NHs before the conjugation of them with the NPs. Lee et al. activated DOTA with EDC/NHs at a pH 5.5 for 30 min in a molar ratio of 10:5:4 (DOTA:EDC:Sulpho-NHS). This mixture was added into the IONPs solution at a pH of 8.5, and it was incubated for 1 h at 4 °C. At the end, ^64^Cu was added to the solution, and this was following the incubation of it of 45 min at 45 °C, and then, it was purified using a PD-10 column [25].

Additionally, the QDs were radiolabelled with ^64^Cu using a previously activated-DOTA. Briefly, DOTA was activated by EDC and sulpho-NHS at a pH 5.5 for 30min in a molar ratio of 10:5:4 (DOTA:EDC:SNHS). Then, the activated DOTA along with NHS–MAL, which is a heterobifunctional linker, were added into a QDs solution at a pH of 8.5. The DOTA–QDs were then radiolabelled by the addition of ^64^Cu in a sodium acetate buffer and incubated at 40 °C for 45 min. The final mixture was purified using a PD-10 column with a RCY that was greater than 90% [26]. Gold nanoshells (NSs), which are used as diagnostic tool for the imaging of neck squamous cell carcinoma in murine models, were radiolabelled following this method: first, the p-NH_2_-Bn-DOTA was conjugated to bifunctional OPSS-PEG2K-NHS in a 1:1 molar ratio and it was incubated overnight at room temperature. The resulting solution was then added to a NS solution in a 10.000:1 molar ratio, and this was followed by it overnight incubation at room temperature on a shaker. ^64^CuCl_2_ was diluted in an ammonium citrate buffer, and it was added to DOTA-NS solution and incubated at 37 °C for 90 min, and this was followed by the addition of a blocking agent, PEG5K-SH, in a 300,000:1 molar ratio and its incubation at room temperature on a shaker for 1 h. The radiolabelled compound, after its purification, showed an LE of 81% [27]. Rossin et al. radiolabelled latex bead NPs that were coated with an anti-ICAM antibody which was previously conjugated to DOTA. Briefly, DOTA and IgG were mixed in Na_2_HPO_4_ at 4 °C overnight (pH 7.5), and they were separated from the excess reagent by their filtration. Following this, ^64^CuCl_2_ was incubated with IgG-DOTA and incubated for 1 h without its further purification being performed. Finally, the latex bead NPs were coated with ^64^Cu-DOTA-IgG/anti-ICAM-1 or ^64^Cu-DOTAIgG/IgG for 1 h at room temperature. The unbound proteins and ^64^Cu-DOTA were removed by a centrifugation procedure (4 min, 12.000 rpm), and the final LE was at approximately 75% [28].

Alternatively, NOTA was often used as a suitable chelator for ^64^Cu.

The IONPs were conjugated with thiol-functionalized NOTA (NOTA-SH). NOTA-SH was prepared by a reaction between the amino group of 2-aminoethanethiol hydrocholoride in the presence of triethanol-amine and the NCS group of p-SCN-Bn-NOTA.

The reaction was incubated at room temperature for 3 h in an N_2_ atmosphere. Subsequently, the NOTA-SH solution was added into the water solution containing the PEGylated DOX-conjugated superparamagnetic iron oxide NPs (SPIONPs). This reaction occurred in a water solution at room temperature for 5 h in an N_2_ atmosphere. When the reaction was completed, the solution was purified by its dialysis for 2 days. The radiolabelling process was performed by adding ^64^CuCl_2_ (which had been previously diluted in a sodium acetate buffer) in the solution containing the functionalized SPIONPs, and it was incubated for 40 min at 40 °C in a constant shaking condition. [29].

Graphene oxide-iron oxide NPs were conjugated to NOTA in a ratio of 1:4 (NPs: chelator), and the reaction occurred overnight before the desalting purification procedure was conducted. ^64^CuCl_2_ was added, and the solution was incubated at 37 °C for 30 min in a constant stirring condition. Finally, the radiolabelled NPs were purified using PD-10 columns with PBS being the mobile phase [30].

For the radiolabelling of Mn_3_O_4_ NPs, Zhu et al. added NOTA-NHS to the Mn_3_O_4_ NPs solution and it was stirred continuously for 24 h. The ^64^Cu labelling was performed at room temperature by adding ^64^CuCl_2_ and incubating it with an ammonium acetate buffer for 15 min. The resulting mixture was then added to the solution of NOTA-Mn_3_O_4_ NPs. After 30 min of incubation, the ^64^Cu–NOTA-Mn_3_O_4_ NPs solution was purified using a PD-10 desalting column [31].

A molar ratio of 1:10 was adopted in the reaction between the zinc oxide NPs (ZnO-NPs) and NOTA at a pH 8.5, which occurred after it was incubated for 2 h. To purify the resulting NOTA-ZnO-NPs, the authors used filters with a cutoff of 100 KDa. After a dilution of ^64^Cu in 0.1 M sodium acetate buffer, ^64^Cu was incubated with the NOTA-ZnO-NPs in a stirring condition (350 rpm) at 37 °C for 30 min. The purification was performed by the filters with a molecular weight cut-off of 50 kDa, and it was finally resuspended in PBS [32].

DTPA is largely used as a BFC for several isotopes, but it is not often applied for chelating ^64^Cu. The DTPA-cross linked IONPs were labelled with ^64^CuCl_2_ in an ammonium acetate buffer at 95 °C for 1 h, thereby obtaining a final RCY of 72% [33].

#### 2.1.3. Discussion

In summary, direct labelling with ^64^Cu is a fast and efficient method, but it is not applicable for all of the types of NPs. Depending on their chemical characteristics, they can have a strong and stable bond or a very weak bond with the radioisotope. To overcome this problem, some authors have suggested a functionalization of the NPs with sulfur or oxygen groups to form a more stable bond and avoid the dissociation between the radioisotope and the NPs.

In the case of indirect labelling, several BFCs have been proposed, each with different characteristics. Most of these have a short incubation time, thereby making this process fast and efficient. However, several authors have seen that the functionalization that occurs with these chelators can affect the properties of the NPs and reduce their targeting specificity [34].

The advantages and disadvantages of ^64^Cu labelling are summarized in Table 1.

### 2.2. Radiolabelling with Gallium-68

^68^Ga is a generator-produced isotope with a relatively low cost when it is compared to the cyclotron-produced isotopes. Despite it achieving non-excellent spatial resolution imaging in PET due to the high energy of positrons on it and its very short half-life (68 min), ^68^Ga is a promising isotope for NP radiolabelling. Like ^64^Cu, ^68^Ga can be radiolabelled either directly or indirectly with a chelating agent, such as DOTA, NOTA, NODAGA, or other BFCs that create a very stable complex with gallium (III)-cation [35]. The widely used purification methods for ^68^Ga-NPs are based on solid-phase extraction (SPE) or size-exclusion chromatography (SEC). However, other methods such as ultracentrifugation have also been applied [36].

#### 2.2.1. Direct Radiolabelling

The QDs with ZnS cores and a PEG-OCH_3_ coating (QD-OCH_3_) were radiolabelled with ^68^Ga through a cation exchange at nearly room temperature in an aqueous solution, thereby obtaining a very high LE. The QDs were doped with ^68^Ga by incubating ^68^GaCl_3_ in a sodium acetate buffer for 15 min at 37 °C. The NPs can subsequently be functionalized with peptides to improve their specificity [37].

Magnetite NPs (Fe_3_O_4_ MNPs) were radiolabelled without a chelator by adding a solution of sodium citrate and ^68^GaCl_3_ and incubating them at 90 °C for 40 min. Before purification, the RCY was ∼70%, as determined by radio-ITLC analysis, but after the purification, the sample showed a radiochemical purity >91% [38]. Another strategy for radiolabelling without the use of BFCs, is the core-doping of the NPs with a radioisotope using microwave-assisted heating. This method has several advantages, such as a reduced reaction time in comparison to the traditional methods, a high reproducibility, and a high LE and yield [39].

Pellico et al. radiolabelled the IONPs used this method by combining FeCl_3_ and dextran (to ensure a colloidal stability) with the generator eluate ^68^GaCl_3_ and heating the mixture to 100 °C (in 54 **s**) with microwave irradiation at 240 W for 10 min. This method turned out to be very efficient and reproducible with a high RCY, and after the purification, this was of 93.4 ± 1.8 [40].

Ligand anchoring group-mediated radiolabeling (LAGMERAL) has been demonstrated to be an efficient strategy for labeling Fe_3_O_4_ NPs. These were initially labelled with ^99m^Tc as proof of concept, and then, they were labelled with ^68^Ga. This method is based on the interaction between the metal radioisotope and the diphosphonate anchoring groups of the PEG-coated NPs [41,42].

#### 2.2.2. Radiolabelling with Bifunctional Chelators

PEG-modified nano-graphene sheets were conjugated with NOTA and functionalized with a TRC105 antibody for the in vivo targeting of the early stages of many tumors. In this study, NOTA was firstly attached to the NPs by binding them to PEG molecules, and this step was followed by the addition of ^66^Ga and its incubation for 30 min at 37 °C under a constant stirring condition [43]. ^66^Ga is an equivalent of ^68^Ga for PET use, but it has a physical half-life of 9 h, which makes more suitable for the pre-clinical kinetic studies.

Cobalt ferrite magnetic NPs that are functionalized with an aptamer-targeting under-glycosylated mucin-1 (uMUC-1) were firstly conjugated with NOTA in an NaHCO_3_ buffer solution while it was vortexed and mildly stirred at 4 °C, and then, radiolabelled with the ^68^Ga. The reaction mixture was incubated for 1 h after it was briefly vortexed for up to 24 h, and it had a high stability [44].

The IONPs were also radiolabelled with NOTA. NOTA was added into the IONPs solution and mixed for 2 h. The reaction mixture was then washed, and finally, it was purified using a PD-10 column [45].

The BFC DOTA was used for the labelling of polyamide dendrimers (PAMAM) that were conjugated with αʋβƷ receptors for the detection of tumor angiogenesis in mouse models with Ehrlich’s ascites tumors (EAT). The conjugation occurred with the addition of a DOTA-NHS ester to the dendrimer’s solution. The mixture was stirred at room temperature for 48 h, and subsequently, ^68^Ga was added in the solution. The reaction mixture was stirred and incubated at 90–100 °C for 15–30 min [46].

Hajiramezanali et al. conjugated SPIONs with N,N,N-trimethyl chitosan (TMC)-coated magnetic nanoparticles (MNPs). The conjugation with DOTA was performed using the amine groups of TMC on the surface of the NPs. It was possible to purify the final solution by centrifugating it because the functionalized NPs were precipitated. The radiolabelling procedure with ^68^Ga was allowed by adding a ^68^GaCl_3_ solution that had been previously eluted with 0.2 M HCl. The mixture was vortexed for 10 **s** and heated at 90 °C for 5 min. This method was very efficient, and it showed a radiochemical purity that was higher than 98% and a stability, in vitro in the human serum, of 92% after 120 min and of 86% after 180 min [47].

The radiolabelling of porous zirconia (ZrO_2_) NPs was performed using DOTA as BFC, which was successfully adsorbed on the surface of the NPs. ^68^Ga-radiolabelling was performed by mixing the DOTA-ZrO_2_ solution with ^68^Ga that had been previously preconditioned using AG 1-X8 resin columns at 95 °C and at a pH 4 for 20 min [48].

NODAGA is another chelator that can be used for the labelling of NPs with ^68^Ga. AGuIX NPs are ultrasmall rigid NPs (5 nm) that are made of polysiloxane and surrounded by gadolinium chelates. Due to their size, they are sufficiently small to escape hepatic clearance. They were functionalized with NODAGA for the following radiolabelling process with ^68^Ga to be performed. The labelling between the NPs and the BFC occurred by dissolving the NODAGA in DMSO, and then, it was gradually added to the AGuIX solution under a stirring condition for 5 h at room temperature. The in vivo studies showed that these NPs remain unmetabolized up to at least 60 min post-injection, thereby making them an excellent imaging agent with there being passive accumulation in the diseased area [49].

The NODAGA was used also by Lahooti et al. for the radiolabelling of ultra-small superparamagnetic iron-oxide nanoparticles (USPION) [50] and by Körhegyi et al. for the labelling of chitosan and poly-glycolic acid (PGA) NPs. In particular, the NODAGA-NHS solution, which had been previously prepared, was added in a dropwise manner to a chitosan solution, and the reaction mixture was stirred at room temperature for 24 h. The chitosan–NODAGA conjugate (CHI-NODAGA) was purified by a dialysis procedure and after the synthesis of folate-labelled PGA, the stable self-assembling NPs were produced via an ionotropic gelation process between PGA-PEG-FA and the CHI-NODAGA conjugate under a continuous stirring condition at room temperature to give an aqueous solution of the conjugated NPs. The radiolabelling was then performed by adding ^68^Ga into the solution and incubating it at room temperature for 15 min [51].

Hydrophilic superparamagnetic maghemite NPs, which were coated with a lipophilic organic ligand and entrapped into polymeric NPs that are made of biodegradable poly(lactic-co-glycolic acid) (PLGA) which is linked to PEG were conjugated on their surface with NODAGA through a classic peptide bond. The purification was carried out by filtering the solution. After the conjugation with NODAGA was achieved, the ^68^Ga eluate was added to the vial, and it was heated at 60 °C for 30 min [36].

Papadopoulou et al. compared the radiolabelling of magnetic iron oxide NPs (MIONs) with two different approaches: one was a chelator-free method, in which the radiolabelling process consisted of the incubation of a mixture of ^68^GaCl_3_ eluate and NPs at 70 °C with pH 4. The ^68^Ga was directly incorporated on the surface of the MIONs due to its affinity of the carboxylic groups of the copolymer coating. The second strategy was the chelator-mediated radiolabelling with NODAGA. The conjugation process was allowed by the bound between the three nitrogen atoms of the macrocyclic ring and the three oxygen atoms of the carboxylate groups of chelators. Both of these approaches were efficient, but the radiochemical purity and RCY of the first approach was better for these nanostructures [52].

#### 2.2.3. Discussion

In conclusion, as shown in Table 2, and similarly to ^64^Cu, the direct labelling methods that are reported in the literature are very fast and they avoid the further manipulations of the NPs. However, high temperatures are often required to achieve a high LE, and this can have an impact on the NPs’ characteristics. The pH is also important, in particular, many authors suggest that other should work in a pH range that is between 3 and 5.

In the indirect method, different chelators were used for radiolabelling the NPs with ^68^Ga thanks to the favorable chemical characteristics of this isotope. The main limitation of this approach is the possible dissociation of the radioisotope from the NPs in the bloodstream due to a transchelation reaction occurring. Now, there are pre-formulated kits that are commercially available that make this type of labelling occur quickly and without the need for further purification steps.

### 2.3. Radiolabelling with Zirconium-89

Metallic radionuclides are excellent candidates for PET applications. ^89^Zr, thanks to its half-life of 3.3 days, has been successfully used with many biomolecules that have long circulation times, such as the antibodies for immuno-PET applications. Similarly, the NPs that have a log-plasmatic half-life may benefit from being labelled with this radioisotope.

#### 2.3.1. Direct Radiolabelling

The direct labelling with ^89^Zr can be performed by using the chemical affinity between the isotope and the NP. In the literature, among the most significant results, the silica based-nanomaterials are often easily radiolabelled with several isotopes due to the affinity of the silanol groups with the oxophilic cations [14]. Indeed, the radiolabelling of the silica NPs with ^89^Zr is possible thanks to the strong interaction between the hard Lewis base (deprotonated silanol groups) and the hard Lewis acid (^89^Zr^4+^). Chen et al. used the favorable characteristics of the radiolabeled ultrasmall cRGDY-conjugated fluorescent silica NPs (C’ dots) to radiolabel them with ^89^Zr. As it is underlined as in this approach, is important to consider the pH and the temperature of the reaction, which should be between pH 8–9 and 50–75 °C, respectively. Indeed, a decrease in the pH (2–3) leads to a protonation of the silanol groups that cannot bind the positively charged ^89^Zr.

Interestingly, they also compared the chelator-free approach with a chelator-based radiolabelling method using DFO as the BFC. The ultrasmall silica NPs (6–7 nm) were radiolabelled, and this achieved a high RCY and 99% stability in the serum at 37 °C for both of these methods. However, the biodistribution studies in vivo showed a higher stability for the non-chelator approach, with there being an increased bone uptake, thereby confirming the detachment of the isotope from the NPs [53].

A similar approach was used to radiolabel manganese-based NPs (Mn_3_O_4_) in a chelator free-way due to the formation of Zr complexes which bind the vacant tetrahedral sites on the surface of the NPs. The radiolabelling was performed by simply mixing the water-soluble Mn_3_O_4_ -PEG with ^89^Zr^4+^ in an HEPES buffer at pH 7–8. The process relied on the temperature (25–75 °C), the incubation time (0–200 min), and the concentration of the NPs (1 × 10^−^^2^–1 mg/mL), thereby obtaining a higher RCY by increasing each of these parameters [54].

The same authors also successfully radiolabelled Gd_2_O_2_S:Eu NPs, exploiting the presence of the oxygen donors of the NPs’ surface. The RCY was over 75% after 30 min of incubation and this result was maintained for 180 min at a high temperature and pH 7–8. A significative difference in the RCY was observed by changing the pH conditions. Indeed, at a pH 2–3, the RCY decreased to 15% due to the protonation of the surface oxygen donors [55].

Fairclough et al. used the chelator-free method for the radiolabelling of chitosan NPs due to the presence of free amino and hydroxyl groups on their surface that allow the conjugation of them with metal ions. Indeed, the radiolabelling process occurred by simply adding the radioisotope in the NPs solution during its incubation in a thermos shaker at 1400 rpm for up to 45 min. After that, the final mixture was centrifuged at 11.600× *g* for 10 min to separate the free-^89^Zr from the ^89^Zr-loaded CNs. To evaluate the final LE, the radioactivity in the supernatants and pellet was counted, and this process obtaining an LE of more than 70% [56,57].

Dextran-coated superparamagnetic iron oxide nanoparticles were directly radiolabelled simply incubating the radioisotope with the NPs for 1 h at 100 °C at a pH 8. The authors noticed that this method caused an increase of the hydrodynamic diameter (from 56 nm to 127 nm) [58].

Another strategy that has been used is the radiochemical doping one, which consists of the adding of the radionuclide to the solution with the nanomaterial precursors, thereby triggering a co-precipitation with the incorporation of the radionuclide [2]. This strategy was used by Chen et al. for the production of ^89^Zr-UiO-66. ^89^Zr-UiO-66 is a nanoscale metal–organic framework (nMOF), which is an interesting tool that can be used for drug delivery or as an imaging probe. The incorporation of the radionuclide occurred during the UiO-66 synthesis. Briefly, as a first step, HCl was added int ^89^Zr-oxalate solution and incubated at 200 °C for 2 h to vaporize all of the oxalates. The obtained solution was added into the reaction system, which has been previously prepared. The reaction system was obtained by mixing zirconium chloride (ZrCl_4_), terephthalic acid (BDC), benzoic acid, and HCl which had been dissolved in DMF at room temperature. Once cooled, the obtained white UiO-66 powder was washed and dispersed in DMF under a stirring condition for 6 h to remove the excess BDC. As final step, acetone was used to disperse UiO-66 and to exchange the trapped DMF. At the end, after a drying process and under a vacuum condition at 60 °C that occurred overnight, the final product was obtained. This method showed a high stability of ^89^Zr-UiO-66, which was evaluated after 120 h of incubation in mouse serum [59].

#### 2.3.2. Radiolabelling with Bifunctional Chelators

DFO is a cyclic hexadentate chelator that is widely used to chelate ^89^Zr. Compared to DTPA, DFO shows a greater stability in vivo, without affecting the in vivo biodistribution of the NPs, and allowing a radiolabelling process to be performed at mild temperatures and with a neutral pH [60,61,62].

The radiolabelling via the ^89^Zr-DFO coupling method usually provides a first step, whereby the NPs are conjugated to DFO, and this is followed by the addition of the isotope.

The DFO can also be used to stably label the isotope in the core of the NPs. For example, Li et al. radiolabelled liposomal NPs with the ligand-exchange method. The authors labelled the 8-HQ (oxine) to the isotope, thereby allowing the delivering of ^89^Zr into the liposomal cavity where it was previously encapsulated in the DFO. Briefly, the authors added the DFO into the NPs solution, and this was followed by 30 min of incubation at 35 °C and 5 min of sonication, thereby allowing the encapsulation of DFO into the liposomal cavity. Then, the radioisotope was chelated with 8-HQ (oxine). The final mixture was kept at room temperature for 30 min before the addition of the DFO-liposome solution, which was followed by another 60 min of incubation. The volume ratio of the final solution of ^89^Zr:8-HQ:DFO-liposome was 2:1:3. The RCY was 98%, but after its storage for 48 h at 4 °C, this was reduced to 83% [63].

Ferumoxytol (superparamagnetic iron oxide NPs that are coated with polyglucose sorbitol carboxymethylether) was labelled with ^89^Zr-DFO, which was used as a PET/MRI contrast agent. For the success of the radiolabelling process, a modification of the surface chemistry of the drug was needed and, in particular, an amination of the particles to bind the DFO to Ferumoxytol was carried out.

After the radiolabelling process, which consisted of adding ^89^Zr in the modified ferumoxytol and mixing them at 37 °C for 1 h, they analyzed the LE before its purification (>90%) and the radiochemical purity (99%, and this remained stable for over 24 h at 37 °C in mouse serum) [64].

High-density lipoprotein (HDL) has been radiolabelled with a high efficiency in several studies, and it is usually applied to image tumor-associated macrophages (TAMs) or activated macrophages in atherosclerosis. The ^89^Zr physical half-life matches the biologic half-life of HDL, thus making ^89^Zr-HDL a perfect radiopharmaceutical. For these studies, the labelling process required a previous modification of HDL with a DFO. The conjugation was obtained via a reaction between the DFO and the lysine amino group of ApoA-1. This method showed a high radiochemical purity [65,66,67,68,69,70,71].

Dextran nanoparticles were studied as a nuclear probe for the detection of inflammatory leukocytes in atherosclerotic plaque. Before the radiolabelling was performed, the NPs were modified with epichlorohydrin through a cross-link reaction, and then, they were aminated with an ethylene diamine, thereby obtaining amino-dextran NPs (DNP-NH2). Finally, they were conjugated with p-isothiocyanatobenzyl desferoxamine (SCN-Bz-Df) and radiolabelled with ^89^Zr, and then, they were added to the final mixture at room temperature [72].

AuNPs were radiolabelled with ^89^Zr and conjugated with a monoclonal antibody (cetuximab) for to test their quantitative imaging performance in a PET application. The monoclonal antibody was first radiolabelled with ^89^Zr via desferal moiety, and then, it was conjugated with AuNPs using carbodiimide chemistry. The radiochemical purity after the purification was >95%. The immuno-PET showed a higher tumor-to-background ratio of ^89^Zr-cetuximab-AuNPs than ^89^Zr-cetuximab did alone, without there being significant differences in the biodistribution, thereby proving that it is a promising tool for a future theragnostic approach. In another study that was conducted by the same group, AuNPs were conjugated with the anti-CD105 antibody which had been previously radiolabelled with ^89^Zr using the same strategy. These NPs were used to perform a quantitative PET imaging of mice bearing tumors. The results confirmed its high specificity in vivo [73,74].

#### 2.3.3. Discussion

As mentioned above, and summarized in Table 3, direct labeling with ^89^Zr is applicable to many types of NPs. Although we can achieve good labelling results, the direct methods require there to be a rigid Lewis base on the surface of the NPs.

Unlike the other radioisotopes, for indirect labeling with ^89^Zr, only DFO has been proposed as a chelator. DFO does not affect the chemical characteristics of the NPs, but it can increase their size; a parameter that can influence their final biodistribution.

### 2.4. Radiolabelling with Iodine-124

Among the positron-emitting radionuclides, iodine-124 (^124^I) has the longest half-life (T_1/2_ = 4.2 days). This characteristic, when it is combined with its chemical properties, contribute to its wide diffusion in the study of NPs pharmacokinetic [75].

There are few data that are available in the literature regarding direct labelling, such as the remote loading method or the use of Iodo-beads and Iodogen, or via Chloramine-T oxidation. On the contrary, for indirect labelling, various techniques have been proposed, including the use of Bolton–Hunter reagent as BFC. Some of these techniques reach the best performing at high temperatures, which can be a limit of them.

#### 2.4.1. Direct Radiolabelling

For the iodine radiolabelling of liposomal NPs, the direct labelling method demonstrated to have a higher efficiency than the indirect method using the Bolton–Hunter reagent did [76,77]. For this reason, a direct method to encapsulate ^124^I in the liposomal NPs has been used. Here, isotopes are conjugated with compounds that allow the passive crossing of them through the membrane of the NPs. The most frequently used compound for the remote loading of ^124^I in the liposomal NPs is the amino diatrizoic acid (ADA), a iodinated contrast agent that is usually applied in Computed Tomography (CT). The compound is first conjugated to the isotope, and then, thanks to solutions that are based on citrate or ammonium sulphate that create a transmembrane pH gradient, the compound is able to cross the lipid membrane. The non-protonated compound, once it is inside the liposomal NPs, is protonated and cannot be released from the inner core [78].

A novel class of NPs, which are defined as “upconversion NPs (UCNPs)”, are composed by fluorescent metal-based materials such as NaYF_4_, NaGdF_4_, NaLaF_4_, LaF_3_, GdF_3_, CeO_2_, LiNaF_4_, etc. They are characterized by an emission in the near-infrared (NIR) spectrum, thus resulting in a high degree of the penetration of the light through the biological tissues [79,80]. Lee et al. combined the optical properties of Er^3+^/Yb^3+^ which was co-doped NaGdF_4_ NPs using PET/MRI property imaging, thereby developing a multimodal tool for tumor angiogenesis imaging. The UCNPs were radiolabelled with ^124^I using Iodo-Beads. The NPs that were functionalized with the arginine-glycine-aspartic acid (RGD) motifs had a surface-exposed tyrosine residue that allowed the direct conjugation of them with ^124^I using the polystyrene beads. The resulting radiolabelling yield was approximately 19%, and the in vivo tumor uptake of ^124^I-c(RGDyk)2-UCNPs was ∼2%ID/g at 4 h, thus confirming that there was radiolabelling instability due to the de-iodination of radioiodine from the NPs. Further studies are needed to improve the stability of radiolabelling [81].

The same method was applied for polymeric NPs that were synthetized by poly(4-vinylphenol) (PVPh) polymers. The large number of phenolic groups on their polymeric backbone allowed an easy radio-iodination to occur, thus resulting in a high radiolabelling yield (~90%). The PVPh-NPs were incubated with iodination beads (Iodo-beads) including the ^124^I isotope. When the beads were removed, the reaction was stopped. The NPs were then conjugated with three different mAbs: anti-adhesion molecule of platelet-1 endothelial cells (PECAM-1), anti-thrombomodulin (TM), and anti-PV1. The results showed that the NPs targeting PECAM-1 enabled a high-quality PET image to be obtained of the pulmonary vascularity in the murine models [82]. A similar approach was used with Iodination vials (Iodogen), where the iodine nuclides are blocked in the reaction vials. The isotope covalently labels the tyrosine motifs on the NPs’ surface [83].

By contrast, the Chloramine-T method has been used to radiolabel Gold NPs. Iodination was performed by adding the Chloramine-T reagent to the solution containing the isotope and the NPs. The free isotope was then removed by ultracentrifugation. The ^124^I-AuNPs were used for in vivo tumor imaging through a micro-PET in a breast cancer mice model and to track the trafficking of the dendritic cells to evaluate the efficacy of the DC-based immunotherapy [84,85].

It has been reported that iodine isotopes have a high affinity for gold nanomaterials, thus resulting in them having a direct and strong bond with them [86]. ^124^I-labeled gold nanostar probes (^124^I-GNS) that are used for brain tumor imaging are selectively brain-tumor-targeting thanks to the EPR effect, thus making the ^124^I-GNS nanoprobe promising for its future clinical applications to diagnose brain tumors [87].

#### 2.4.2. Radiolabelling with Bifunctional Chelators

The Bolton–Hunter method has been successfully used to radiolabel silica NPs with ^124^I. The NPs with an average diameter of 20–25 nm and surface-free amino groups were efficiently conjugated with a covalent linkage to the NHS ester group that had been previously radiolabelled with ^124^I (^124^I-NHS) for the PET imaging to be performed in vivo [88].

#### 2.4.3. Discussion

Iodine isotopes have a high affinity for the aromatic rings that are present on the tyrosine residues, and their presence is sufficient to promote Iodine oxidation to induce the direct labelling of them to several molecules. This oxidation process must be followed by a reduction one with a reducing agent. There are fast and efficient methods for inducing iodine oxidation, the most commonly used one being Chloramine-T or N-bromo-succinimide. To avoid the following reduction, the oxidative reaction has been coupled with beads (Iodo-beads) or to a reagent that is insoluble in water (Iodogen). These methods are slower, but they are gentler and usually preserve the structure of the NPs. By contrast, very few reports are present in the literature using the Bolton–Hunter reagent as BFC agent for NP labelling. The published experience with ^124^I is summarized in Table 4.

### 2.5. Radiolabelling with Fluorine-18

Fluorine-18 that is labelled with a deoxyglucose molecule ([^18^F]-FDG) is the main radiopharmaceutical that is used in clinical PET imaging. The main drawback of this radionuclide is its short half-life (T_1/2_ = 109.7 min), which restricts its use to studies of small molecules with a fast biodistribution. The NPs generally have a longer pharmacokinetic that does not match with the half-life of this isotope, thus limiting its use in nanomedicine.

#### 2.5.1. Direct Radiolabelling

One strategy for directly radiolabelling the NPs with ^18^F is based on bombarding the nanomaterials with a neutron or proton, whereby an atom of the NP undergoes a nuclear reaction, thereby providing a radionuclide in situ. This strategy was applied for the radiolabelling of ^18^O-enriched tin oxide (Al_2_O_3_) NPs by their direct irradiation with 16 MeV protons. The nuclear reaction allowed the transmutation of ^18^O in ^18^F. This method provided the precise control of the isotope position, thus achieving a high radiochemical stability. However, its application is limited to inorganic nanomaterials since the organic NPs can be affected and modified in their structure by the nuclear reaction. Furthermore, this method requires specific instrumentation with a high management costs [89]. Unlike the metal radionuclides that prefer to undergo labelling via chelators, the halogen radionuclides, such as ^18^F, are usually labelled directly with a chemical group (chemical adsorption) or with a prosthetic group (indirect labelling) on the surface of the NPs. Chemical adsorption usually occurs with the reaction between the soft acids and the soft bases or between the hard acids and the hard bases, thereby creating strong coordination bonds between the isotope and chemical groups on nanomaterials. Several studies have been reported in the literature, showed the strong affinity between ^18^F and the rare earth NPs, such as KGdF_4_, NaYF_4_:Yb, Gd-NaYF_4_:Yb, NaYF_4_:Yb, and NaYF_4_:GdYb. The chemical adsorption of fluorine on the NPs’ surface is a simple and fast method, whereby only the incubation of the isotope with the NP leads to a chemical stability of the compound with a RCY that is higher than 90% and a high radiochemical stability in vivo. The main limitation of this approach is the high temperatures that are required to achieve the conjugation [90,91,92].

Rare-earth fluoride NPs, such as yttrium trifluoride (YF_3_) nanoparticles could be radiolabelled by mixing [^18^F] the potassium fluoride solution with an aqueous solutions of NPs at room temperature, which would be followed by a 5 to 10 min incubation procedure. The free ^18^F can then be removed by centrifugation. Excellent radiolabelling yields were reported, which were in the range of 80–95% [93]. This strategy could be also used with magnetic nanoparticles, such as MnFe_2_O_4_ and Fe_3_O_4_, where the radiolabelling process consists of adding a [^18^F] sodium fluoride solution in a solution of NPs and incubating them while they are continuously shaken at room temperature for 10 min [94]. Indeed, UCNPs that are composed of lanthanide nanocrystals (Gd^3+^/Yb^3+^/Er^3+^) with co-doped NaYF_4_ were efficiently and directly radiolabelled with ^18^F through a simple incubation. The strong binding between Y^3+^ and F^−^ allowed for a high LE. In vivo, the low bone uptake demonstrated the stability of this radiopharmaceutical.

The advantage of lanthanide materials is that they are characterized by their luminescent and magnetic properties, which provide a high spatial resolution and a high sensitivity when they are used in MRIs and fluorescent imaging, while the positron-emitting radionuclide provides functional information in PET imaging. Indeed, with a single nano-radiopharmaceutical, it is possible to obtain multimodal imaging at the molecular level with high sensitivity [95].

#### 2.5.2. Radiolabelling with Bifunctional Chelators

For the indirect surface labelling of ^18^F with prosthetic groups, it is typical that the copper-catalyzed azide–alkyne cycloaddition click chemistry is applied [96]. With this method, the prosthetic groups of the but-3-yn-1-amine modified USPIONPs, maleimide-AuNPs, and aminated IONPs were efficiently conjugated with ^18^F under mild conditions and with high yields [97,98,99].

Nanodiamonds (DNPs) are sp3-carbon NPs, which are a promising biomaterial due to their good biocompatibility, possibility to be functionalized for drug delivery and ability to cross the cell membrane. The radiolabelling of these NPs was made possible by covalently attaching the ω-aminopropyl groups to the surface of the DNP, a reaction that occurs under mild conditions with high yields and is a well-established methodology for functionalizing various solid materials, including silicas and metal oxides. The resulting amino-DNPs were treated with ^18^F-SFB (N-Succinimidyl 4-[^18^F] Fluorobenzoate), thereby obtaining ^18^F-radiolabelled NPs. In the biodistribution studies, it was observed that these NPs accumulate in the lung, spleen, and liver and are excreted into the urinary tract [100].

[^18^F]-SFB was used also by Guerrero et al. for radiolabelling AuNPs. ^18^F-SFB generally reacts with the primary and secondary amino groups. For this reason, the AuNPs were functionalized with two peptides that can react with the radioactive [^18^F]-SFB moiety, thereby allowing the radiolabelling to be performed [101]. For radiolabelling the AuNPs, a covalent binding strategy is preferable for an in vivo stability. To easily radiolabel the AuNPs with ^18^F, Zhu et al. synthesized, for the first time, an ^18^F-labelled prosthetic group, thiol (4-(di-tert-butylfluorosilanyl) benzenethiol ([^18^F] SiFA-SH, [^18^F]-5), which was labelled through an isotope-exchange reaction with ^18^F in a previous step of the AuNP labelling procedure [98]. Sirianni et al. developed a synthesis method for radiolabelling biotin derivatives to enable the detection of the polymer NPs using a ^18^F-labelled prosthetic group, [^18^F]4-fluorobenzylamine, and commercially available biotin derivatives, NHS-PEGn-Biotin. [^18^F]-NPB4 ([^18^F]-fluorobenzylamide-poly-(ethylene glycol)4-biotin) was then linked to the avidin-modified PLGA NPs, thereby developing a radioligand to facilitate the detection of the avidin-modified polymer NPs in the tissues [102].

Finally, Sun et al. developed a procedure for coating the NPs with a polymeric layer for the easy ^18^F labelling of the IONPs using the bifunctional chelator NOTA to act on the comb-like amphiphilic polymer for the chelation of the aluminum fluoride ions. They started with the preparation of the comb-shaped branched polyacrylic acid (COBP), which was synthesized from polyacrylic acid and polyamine, and after that, they conjugated it with NOTE (NOTE-COBP) via the amide bonds. The synthesized IONPs were coated with a layer of NOTE-COBP molecules via a ligand addition method. Finally, the ^18^F aluminum fluoride ions (^18^F-AlF) were chelated with the NOTE groups on the NOTE-COBP coating of the NPs [91].

#### 2.5.3. Discussion

As summarized in Table 5, direct fluoride labelling requires precise instrumentation, and it is not an easy or cheap technique, but has the advantage to provide a stable bond with the NPs simply by incubating them with the radioisotope. However, this technique cannot be used with all of the types of NPs, and it is limited only to the inorganic materials.

Among the indirect techniques, the use of prosthetic groups seems the most reliable. It requires high temperatures, but it is an easy and fast process, and above all, it minimizes the transchelation.

## 3. General Conclusions

In this review, we highlighted the different methods of radiolabeling the NPs with several isotopes that can be used for diagnostic purposes in PET applications. Different approaches have been discussed depending on the type of NP and the type of radioisotope.

Indeed, the first approach, when one is aiming at developing a new NP-based radiopharmaceutical for PET imaging, is to investigate the chemical characteristics of the NP. Then, the choice of isotope should be taken into consideration according to its chemistry and its physical half-life. When these two parameters have been fully analyzed, the most appropriate method of radiolabelling, either a direct or an indirect method, can be chosen. Each procedure must be followed by accurate in vitro quality controls, which are followed by biological assays and/or pre-clinical studies to investigate the specificity of targeting the desired tissue.

If chemical–physical characteristics of the NPs allow it, it is preferable to choose a direct method of radiolabelling without the use of a chelating agent. This approach, in general, offers the advantage to not requiring subsequent manipulations, thus avoiding possible modifications of the NP and, consequently, of their biodistribution or of their targeting ability.

When direct labeling is not possible, BFCs such as NOTA, DOTA, DFO, DTPA, and NODAGA, or other molecules that favor a stable bond between the NP and the radioisotope, can be used.

Although efficient indirect radiolabeling methods have been discussed, this method usually involves the use of high temperatures that can alter the structure of the NPs, thus representing a limit to their use.

As shown in this review, there are many methods that have been proposed for radiolabelling NPs. It is, therefore, important to standardize these procedures and obtain reproducible protocols in order to translate their use to the clinic.

## Figures and Tables

**Figure 1 biomolecules-12-01517-f001:**
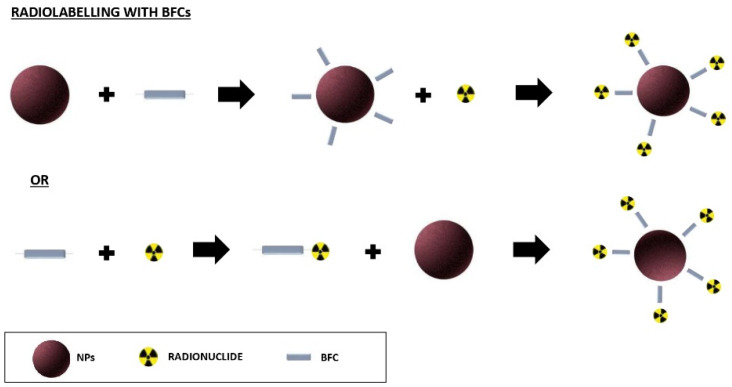
Different approaches to radiolabel NPs using BFCs.

**Table 1 biomolecules-12-01517-t001:** Pros and cons of different methods for radiolabelling NPs with copper-64.

Method/Isotope	Advantages	Disadvantages	Indications	Improvement
Direct labelling with ^64^Cu	Fast and efficient, requires minimal purification	Applied to limited nanoparticles	A previously reduction of ^64^Cu by hydrazine could improve the radiochemical efficacy	Functionalization of NPs with both sulfur (soft) and oxygen (hard) groups to form a stable bond
Indirect labelling with ^64^Cu	Post-synthesis process, allowing the storage of functionalized NPs	Could influence the properties of the nanoparticles and reduce the capability of loading	Efficient radiolabeling with DOTA in short incubation periods requires high temperatures	Increasing the number of chelators bound to each NPs can lead to a decrease in immunoreactivity

**Table 2 biomolecules-12-01517-t002:** Pros and cons of different methods for radiolabelling NPs with gallium-68.

Method/Isotope	Advantages	Disadvantages	Indications	Improvement
Direct labelling with ^68^Ga	Fast one step labeling method	Requires high temperatures	Ammonium acetate is the most suitable buffer solution for labeling process	Optimal pH conditions range from pH 3 to pH 5
Indirect labelling with ^68^Ga	Use of a wide panel of chelators	Different cations in the bloodstream (Ca^2+^ and Mg^2+^) may trigger transchelation, displacing radioisotope in the coordination complex	Chelate ligand and nanoparticle are preferably attached by a covalent bond	Pre-formulated kit with no previous post-processing of the eluate or further purification of the final product

**Table 3 biomolecules-12-01517-t003:** Pros and cons of different methods for radiolabelling NPs with zirconium-89.

Method/Isotope	Advantages	Disadvantages	Indications	Improvement
Direct labelling with ^89^Zr	Applicable to various types of NPs	Needs a hard Lewis base on the nanoparticles’ surface	Better to combine it with biomolecules that have long circulation times	
Indirect labelling with ^89^Zr	Not affect the in vitro e in vivo stability	May increase the particle’s hydrodynamic radius	Only the DFO can be used	Few data in the literature are available

**Table 4 biomolecules-12-01517-t004:** Pros and cons of different methods for radiolabelling NPs with 124-iodine.

Method/Isotope	Advantages	Disadvantages	Indications	Improvement
Direct labelling with ^124^I	Few data in the literature are available	Few data in the literature are available	High affinity with gold nanomaterials	Few data in the literature are available
Indirect labelling with ^124^I	There are several ways of radiolabelling	Requires high temperatures	Using NPs with phenolic groups allows radio-iodination	Using NHS could improve the radiochemical efficacy

**Table 5 biomolecules-12-01517-t005:** Pros and cons of different methods for radiolabelling NPs with 18-fluoride.

Method/Isotope	Advantages	Disadvantages	Indications	Improvement
Direct labelling with ^18^F	Only the incubation of the isotope with the radionuclide leads to a chemical stability of the compound	Requires specific instrumentation with a high management cost	Limited to inorganic nanomaterials	Occurs strong coordination bonds between the isotope and chemical groups on nanomaterials
Indirect labelling with ^18^F	Simple and fast method, minimize transchelation	Requires high temperatures	Usually applied the copper-catalyzed azide–alkyne cycloaddition click chemistry	Better to use prosthetic groups for the radiolabelling process

## Data Availability

Not applicable.

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
