# Peer review of "Methods for Radiolabelling Nanoparticles: PET Use (Part 2)"

_biomolecules, 2022, doi:10.3390/biom12101517_

Round 1

Reviewer 1 Report

Corresponding to Part 1, the authors reviewed the methods of radiolabeling nanoparticles with several radionuclides commonly used for PET in this part, i.e., 64Cu, 68Ga, 89Zr, 124I, and 18F. This manuscript has many of the same problems as Part 1, and it is recommended to be published after addressing following major concerns.

1.      The biggest problem is also that most of the content are superficial, and the authors only briefly retell the labeling process, but hardly explain the labeling principle, and many examples have almost no essential difference (e.g., 2.1.2). What is more, section 2.3 is just a simple enumeration. It is recommended to introduce the principle and describe what type of nanoparticles the method is applicable to, so as to facilitate the expansion of application.

2.      The introduction in this manuscript does not seem to present the reader with anything different from that in Part 1. Even the contents in lines 47-49 and 41-42 seem to be the same in current manuscript?

3.      Lack of necessary figures.

4.      The principle of the labeling in line 91-95 should be chemisorption between 64Cu and -SH of amine-PEG-thiol, and the coordination of 64Cu and -COOH of PAA. The authors need to check current statement.

5.      In addition to the two categories introduced by the authors, there are many other methods for 64Cu labeling, such as radiolabeling doping during synthesis (e.g., Nanoscale, 2014, 6, 13501-13509), and cation exchange (e.g., J. Am. Chem. Soc. 2014, 136, 1706-1709).

For 68Ga, like 64Cu, it can also be labeled by radiolabeling doping (e.g., Contrast Media Mol. Imaging, 2016, 11, 203-210). Additionally, metal ions include 68Ga have been demonstrated to be able to label DP-PEG-coated nanoparticles directly by LAGMERAL method (Small, 2021, 17(51), 2104977; ACS Appl. Mater. Interfaces, 2022, 14, 8838-8846).

For 89Zr, labeling can also be achieved by radiolabeling doping (e.g., ACS Nano, 2017, 11, 4315-4327).

All of these methods should not be absent in a comprehensive review, and it is suggested to refer to the review of similar topics (ACS Appl. Nano Mater. 2022, 5, 7, 8680-8709; Chem. Soc. Rev., 2021, 50, 3355-3423; Biomaterials, 2020, 228, 119553; J. Nucl. Med., 2019, 59(3), 382-389).

6.      In lines 202-203, it is necessary to give references to support the statement.

7.      Throughout the manuscript, there are many irregularities in the format and language. At its most basic, there is usually a space between the number and the unit, and the full name should be given when the abbreviation first appears. It is recommended to carefully check, modify and polish the manuscript.

Author Response

File attached

Reviewer 2 Report

The authors report a summary of radiolabeling methods for PET purposes.

The review presents sufficient and clear information on the methodologies described.

Author Response

File attached

Round 2

Reviewer 1 Report

In the revised edition, the authors have made appropriate modifications to the problems raised previously, but there are still some minor problems, such as part of suggested references are still absent, and there remain some language and grammar errors. It is recommended to accept after minor revision.

Author Response

Replies enclosed
